# Cohort profile: the PHARMO Perinatal Research Network (PPRN) in the Netherlands: a population-based mother–child linked cohort

E. Houben [1,2] L. Broeders,[3] E.A.P. Steegers,[2] R.M.C. Herings[1,4]

¹PHARMO Institute for Drug Outcomes Research, Utrecht, The Netherlands
²Department of Obstetrics and Gynaecology, Erasmus MC, Rotterdam, The Netherlands
³Perined, Utrecht, The Netherlands
⁴Department of Epidemiology and Biostatistics, Amsterdam UMC, Amsterdam, The Netherlands

**Correspondence to**
E. Houben; pharmo@pharmo.nl

## ABSTRACT

**Purpose** Observational population-based research is a very suitable non-invasive method for studies in the vulnerable populations of pregnant women and children. Therefore, the PHARMO Perinatal Research Network (PPRN) was set up as a resource for life course perinatal and paediatric research by linking population-based data from existing registrations.

**Participants** From 1999 to 2017, the PPRN captures approximately 542 900 pregnancies of 387 100 mothers ('Pregnancy Cohort'). Additionally, mother–child linkage is currently available for a quarter of these pregnancies ('Child Cohort'). The PPRN contains preconceptional information on maternal healthcare, as well as detailed pregnancy and neonatal data, extending into long-term follow-up and outcomes after birth for both mother and child up to nearly 20 years. It includes linked data from different primary and secondary healthcare settings.

**Findings to date** Through record linkage of the Netherlands Perinatal Registry and the PHARMO Database Network, we have established a large population-based research network including data on demographics, medication use, medical conditions and details concerning labour, birth and neonatal outcomes. Here, we provide an overview of record types available from the PPRN, available database follow-up and pregnancy characteristics of the PPRN cohorts. The PPRN has been used for a number of different pharmacoepidemiological studies, for example, to confirm that preterm-born infants were more likely than full-term infants to be hospitalised or use medication. Similar long-term comparisons showed that children born following spontaneous preterm labour were at increased risk of neurodevelopmental and respiratory conditions. Most recently, the PPRN provided important evidence on the trends in use of potentially harmful medication during pregnancy.

**Future plans** The PPRN provides a unique and rich data set facilitating large-scale observational pharmacoepidemiological perinatal research. The patient-level linkage of many different healthcare data sources allows for long-term follow-up of mother and child, with ongoing annual updates.

## INTRODUCTION

Observational population-based research is a very suitable non-invasive method for studies in the vulnerable populations of pregnant women

## Strengths and limitations of this study

► The main strength of the PHARMO Perinatal Research Network (PPRN) lies in the ongoing assembly of detailed, population-based, anonymised data from existing registrations which makes it an invaluable resource for life course perinatal and paediatric research.

► Evidence from clinical trials is often lacking behind for pregnant women and children, as researchers are often hesitant to include these subjects due to a variety of reasons. Hence, the PPRN provides a very suitable, non-invasive method that stays within the many risks and objections of studies in the vulnerable populations of pregnant women and children.

► The PPRN covers a considerable proportion of pregnancies from 1999 onwards that has been shown to reflect true estimates of the Dutch population captured in Netherlands Perinatal Registry (Perined), ensuring a high level of generalisability.

► Data collection periods and catchment areas vary between the linked databases and therefore the size of the study population depends on the databases included.

► Currently, we rely on probabilistic linkage methods as the number of records that include a social security number is currently too limited to allow for deterministic record linkage between the PHARMO Database Network and Perined.

and children. Therefore, the PHARMO Perinatal Research Network (PPRN) was set up as a resource for life course perinatal and paediatric research by linking population-based data from existing registrations. It was initiated around 2010 at the PHARMO Institute for Drug Outcomes Research in collaboration with Netherlands Perinatal Registry (Perined). At that time it was set up to study the relation between medication exposure during pregnancy and pregnancy outcomes, but the applications of the PPRN have extended considerably over the years, along with the continuous expansion of the underlying databases.

BMJ

## COHORT DESCRIPTION

### Setting

The PPRN is a unique linkage of the Perined and the PHARMO Database Network (PHARMO). With data collection starting in 1999, the linkage of these population-based data sources facilitates large-scale observational pharmacoepidemiological perinatal research. It contains preconceptional information on maternal healthcare extending into long-term follow-up and outcomes after birth for both mother and child, with ongoing annual updates of the routinely collected data.

### Data sources

Perined is a nationwide registry in which medical data around pregnancy and birth are included from pregnancies with a gestational age of at least 16 weeks (including terminated pregnancies and stillborns).[1] It is a linked database combining medical registries from four professional groups that provide perinatal care: general practitioner, midwives, gynaecologists and neonatologists/paediatricians. Among the items reported are maternal demographics and medical conditions, pregnancy complications and details concerning labour, birth and neonatal outcomes. Linking the records is a complex operation—especially when it comes to records that originate from different data sets. Probabilistic linkage based on matching data is performed in the absence of unique identification of mother and/or child. There is a firm basis for deciding whether two records describe the same case or have a lot of resemblance. The threshold value for such a decision depends on the situation and is statistically substantiated.[2] The established registry reflects virtually all deliveries in the Netherlands (~99% agreement with the municipal administration), that is, including home as well as hospital births. The frequency of data collection and processing is four times a year. The average lag time of the data is half a year.

PHARMO is a population-based network of databases combining subnational data from different primary and secondary healthcare settings in the Netherlands. These different data sources, including data from general practices, inpatient and outpatient pharmacies, clinical laboratories, hospitals, the cancer registry, pathology registry and perinatal registry, are linked on a patient level through validated algorithms.[3] Data are retrieved directly from the source, that is, the electronic medical records of the healthcare providers who agree to contribute to PHARMO. All patients registered at the contributing healthcare providers are included, unless the patient requested to opt out. To ensure the privacy of the data in the PHARMO Database Network, the collection, processing, linkage and anonymisation of the data are performed by the foundation 'Stichting Informatievoorziening voor Zorg en Onderzoek' (STIZON). STIZON is an independent ISO/IEC 27001 certified foundation, which acts as a trusted third party (TTP) between the data sources and the PHARMO Institute. Detailed information on the methodology and the validation of the used record linkage method can be found elsewhere.[4 5] PHARMO covers approximately a quarter of the Dutch population and is shown to be representative of the Dutch population with regard to age and sex; however, data collection period, catchment area and overlap between data sources differ. The PHARMO databases are linked on an annual basis, the average lag time of the data is 1 year.

### Perined–PHARMO linkage

STIZON also acts as a TTP for the linkage between Perined and PHARMO. This specific linkage is primarily based on the birth date of the mother and child, their gender and their zip codes. In case multiple possible links are established, these determinants are supplemented with hospital admission records around delivery as well as obstetrician or gynaecologist-prescribed medication. Furthermore, home codes that indicate mother and child live on the same address are used to verify established pairs and improve linkage specificity.

### Data collection

From 1999 to 2017, the PPRN captures approximately 542900 pregnancies of 387100 mothers for which a PHARMO–Perined link could be established (ie, 'Pregnancy Cohort'). Additionally, an individual mother–child linkage is currently available for a quarter of these

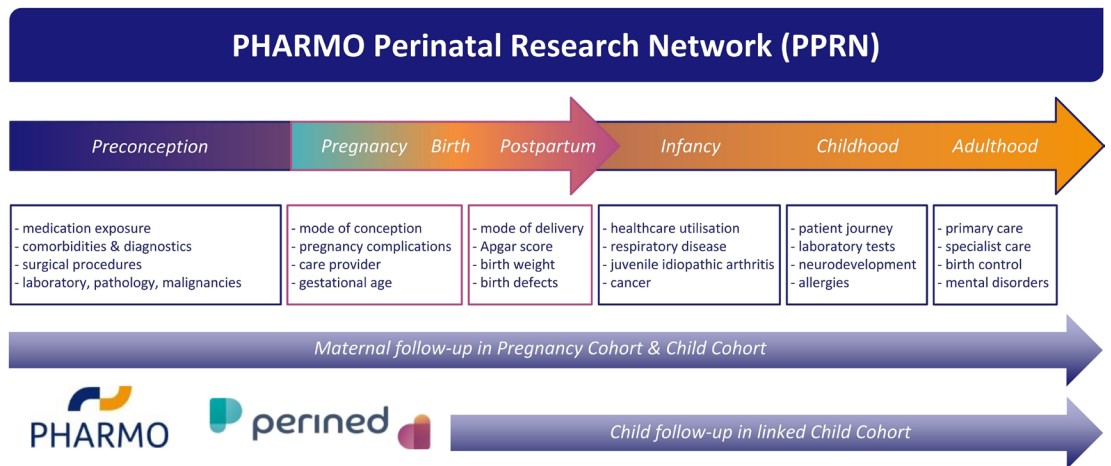

**Figure 1** Schematic overview of data captured in the PHARMO–Perined linked PHARMO Perinatal Research Network (PPRN).

**Table 1** Overview of the record types available from the PPRN

| Record type | Description | Data availability |
|---|---|---|
| Pregnancy/neonatal | Maternal and neonatal characteristics and perinatal care from pregnancies with a gestational age of at least 16 weeks captured by midwife practices, gynaecologists, paediatricians and neonatologists (maintained by Perined). | From 1999 onwards (full coverage for PPRN cohorts). |
| Medication | General practitioner or specialist prescribed healthcare products including information on type of product, date, strength, dosage regimen, route of administration, prescriber specialty and costs dispensed by outpatient pharmacy. Inpatient medication available for a subcohort. | From 1998 onwards (full coverage for PPRN cohorts for outpatient medication; inpatient medication only for a subcohort). |
| General practitioner | Patient records registered by general practitioners (gatekeeper of the Dutch healthcare system) including information on diagnoses and symptoms, laboratory test results, general practitioner visits and referrals to specialists. | From 2003 onwards (partial coverage for PPRN cohorts). |
| Laboratory tests | Results of laboratory tests performed on clinical specimens, requested by general practitioners or medical specialists including information on date and time of testing, test result, unit of measurement and type of clinical specimen. | From 1998 onwards (partial coverage for PPRN cohorts). |
| Hospital admissions | Hospital admissions (ie, inpatient hospital records) for more than 24 hours and admissions for less than 24 hours for which a bed is required including information on hospital admission and discharge dates, discharge diagnoses and procedures (maintained by the Dutch Hospital Data Foundation).[12] | From 1998 onwards (full coverage for PPRN cohorts up to 2015; partial coverage for PPRN cohorts from 2016 onwards) after permission is granted by the Dutch Hospital Data Foundation. |
| Ambulatory care | Ambulatory care (ie, outpatient hospital records; eg, by paediatrician), including diagnosis, number of visits, involved specialist (maintained by the Dutch Hospital Data Foundation).[12] | From 2016 onwards (partial coverage for PPRN cohorts) after permission is granted by the Dutch Hospital Data Foundation. |
| Pathology reports | Data on excerpts of histological, cytological and autopsy examinations, obtained through linkage with the national Pathology Registry[13] (maintained by the PALGA Foundation). | From 1998 onwards (full coverage for PPRN cohorts) after permission is granted by the PALGA Foundation. |
| Malignancies | Data on all newly diagnosed cancer cases including information on cancer diagnosis, tumour staging, tumour site, morphology and initial treatment, obtained through linkage with the national Netherlands Cancer Registry[14] (maintained by the Netherlands Comprehensive Cancer Organisation). | From 1998 onwards (full coverage for PPRN cohorts) after permission is granted by the Netherlands Comprehensive Cancer Organisation. |

PPRN, PHARMO Perinatal Research Network.

pregnancies allowing subjects to be followed over time up to nearly 20 years after birth and studying associations with pregnancy or neonatal-specific outcomes (ie, 'Child Cohort'). A schematic overview of data captured in the PPRN for mothers and children and how these two cohorts inter-relate is included in figure 1 and table 1. Further characterisation of the PPRN is included in table 2, including the total Perined population as a reference, considering that only a subsample of the Netherlands is represented by the PHARMO Database Network. Figure 2 presents the number of pregnancies included in the Pregnancy Cohort and Child Cohort by calendar year. Details on the available database follow-up for the children included in the Child Cohort are presented in figure 3, with end of follow-up defined by either end of database registration (ie, the patient moves out of the PHARMO catchment area), death

or end of study period (31 December 2018), whichever occurred first.

### Patient and public involvement

No patients were involved in the described linkage between existing registries providing an anonymous data set.

### FINDINGS TO DATE

Through record linkage of Perined and PHARMO, we have established a large population-based research network including data on demographics, medication use, medical conditions, pregnancy complications and details concerning labour, birth and neonatal outcomes. The PPRN has been used for a number of different pharmacoepidemiological studies (see online supplemental appendix 1 for a citation list of work published on the PPRN). Its applicability can be centred on the mother, the child or both (ie, assessing

| Table 2 | Pregnancy characteristics in the Pregnancy Cohort and the Child Cohort of the PPRN | | | |
|---|---|---|---|---|
| | | PHARMO Perinatal Research Network (PPRN) 1999–2017 | | Perined 1999–2017 (reference) |
| | | Pregnancy Cohort | Child Cohort | Total population |
| Pregnancies (n) | | ~542 900 | ~126 200 | ~3 200 000 |
| Mothers (n) | | ~387 100 | ~101 400 | – |
| Maternal characteristics | | | | |
| Age at delivery (mean±SD; years) | | 31.0±4.8 | 30.8±4.7 | 31.0±4.9 |
| Nulliparous (%) | | 46 | 57 | 47 |
| Dutch ethnicity (%) | | 79 | 84 | 79 |
| Database history before delivery (mean±SD; years) | | 6.0±4.3 | 6.1±4.3 | – |
| Database follow-up after delivery (mean±SD; years)* | | 7.9±5.0 | 7.7±4.7 | – |
| Infant characteristics | | | | |
| Male sex (%) | | 51 | 53 | 51 |
| Gestational age at birth (mean±SD; weeks) | | 39.1±3.5 | 39.2±2.7 | 39.3±2.3 |
| Preterm birth (%) | | 8 | 7 | 8 |
| Multiple birth (%) | | 2 | <0.5 | 4 |
| Database follow-up after birth (mean±SD; years)* | | 7.8±4.7 | 7.8±4.7 | – |

*Current censoring: 31 December 2018.

the association between maternal characteristics and child outcomes). As an example, medication use during first year of life and hospital admission rates have been assessed and compared between premature and term infants.[6] Preterm-born infants were up to two times more likely than full-term infants to be hospitalised or use medication, especially related to respiratory disease. Similar long-term comparisons of morbidities and healthcare utilisation have been made which showed that children born following spontaneous preterm labour (irrespective of gestational age at delivery) were at increased risk of neurodevelopmental and respiratory conditions compared with those from full-term labour pregnancies.[7] Most recently, data from the PPRN have been used to determine population-based trends over the last two decades in the use of potentially harmful medication among pregnant women.[8]

## STRENGTHS AND LIMITATIONS
The main strength of the PPRN lies in the ongoing assembly of detailed, population-based, anonymised data from existing registrations which makes it an invaluable resource for life course perinatal and paediatric research. Evidence from clinical trials is often lacking behind for pregnant women and children, as researchers are often hesitant to include these subjects due to a variety of reasons, including the fear of harm to the fetus and threat of legal liability.[9 10] Therefore, the PPRN provides a very suitable, non-invasive method that stays within the many risks and objections of studies in the vulnerable populations of pregnant women

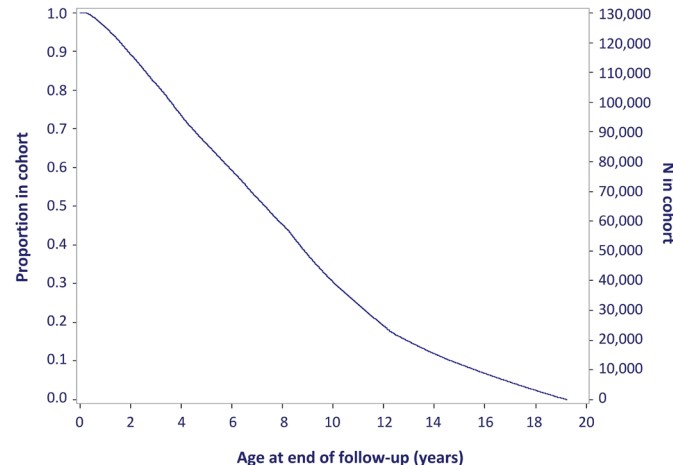

**Figure 3** Proportion and number of children included in the PHARMO Perinatal Research Network (PPRN) Child Cohort, with age in years at end of follow-up (current censoring: 31 December 2018).

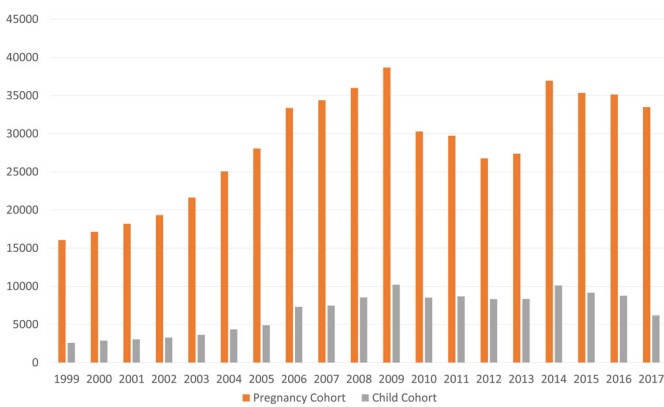

**Figure 2** Number of pregnancies included in the Pregnancy Cohort and Child Cohort by calendar year.

and children. The PPRN covers a considerable proportion of pregnancies from 1999 onwards that has been shown to reflect true estimates of the Dutch population captured in Perined,[11] ensuring a high level of generalisability. The patient-level linkage of many different healthcare data sources provides a very rich data set allowing long-term follow-up of mother and child, with data continuously being collected.

The PPRN brings together data from various sources. Data collection periods and catchment areas vary between these databases and therefore the size of the study population depends on the databases included. The 542 900 pregnancies linked in the data cut up to 2017 allow for assessment of drug use during the 9-month preconception, pregnancy and 9-month postpartum periods. Inclusion of other databases (eg, general practitioner records or hospital admissions) will reduce the cohort size. As with any database, identification of medical events is limited to data that are captured as part of the medical records or other linked data sources in daily clinical practice. These data are not primarily collected for research purposes and rely on appropriate diagnostic coding. Also, the lag time for the PHARMO–Perined linked data to become available is currently approximately 1 year. Furthermore, the number of records that include a social security number is currently too limited to allow for deterministic record linkage between PHARMO and Perined. This availability is steadily increasing and will in the future improve the ability to differentiate between siblings in case of multiple birth pregnancies, which are now under-represented in the Child Cohort. The current linkage methods particularly gain a high specificity, and including these unique patient identifiers the sensitivity is expected to increase further as well. The seeming under-representation of multiple births in the Pregnancy Cohort is caused by the fact that the presented reference proportion for the total Perined population includes all pregnancies (including terminated pregnancies and stillborns); however, comparisons by gestational age indeed show agreement between the two (data not presented). Furthermore, the higher proportion of nulliparous women in the Child Cohort compared with the other two cohorts is mainly influenced by families more often moving houses shortly after delivery of a second child compared with their first child, and due to the new address it is less likely that the child can be traced back in the PHARMO Database Network.

## COLLABORATION

Access to the PPRN is, by governance regulations of the data collection and contractually agreed between the PHARMO Institute and Perined, restricted to researchers of the PHARMO Institute, Perined and academic affiliates. Academic affiliates from universities, hospitals or other research institutes are encouraged to apply for access to the anonymised data for scientific study purposes. The data are handled in accordance with data protection, privacy regulations and ISO certification schemes. Each data request

is checked against these policies and requires permission of the applicable compliance and privacy boards of both PHARMO and Perined. Permission to external databases is requested from the database holders (eg, Dutch Hospital Data Foundation or PALGA Foundation) on a project basis. As it concerns database research with anonymous data, no Institutional Review Board or ethics committee approval is required. An overview of the variables included in the different databases, the terms and conditions and data application forms are available on http://pharmo.nl/what-we-have/data-request-PHARMO/ and (in Dutch) www.perined.nl/registratie/faciliteren-onderzoek. Data sets are processed in SAS version 9.4 (SAS Institute), but can be converted to other data formats. Only a 10% subsample of the requested data can be downloaded by the researcher from a secure FTP server; access to the full data set can be granted to researchers guesting at the PHARMO office.

**Contributors** EH and RH had full access to all of the data in the study and take responsibility for the integrity of the data and the accuracy of the data analysis. EH, LB, EAPS and RH contributed to the plan and design of the study. EH performed the data analyses and drafted the manuscript and was in charge of the study planning. EH, LB, EAPS and RH contributed to the interpretation of the results and critical revision of the manuscript for important intellectual content and approved the final version of the manuscript. EH and RH are the guarantors of this paper.

**Funding** The authors have not declared a specific grant for this research from any funding agency in the public, commercial or not-for-profit sectors.

**Competing interests** None declared.

**Patient and public involvement** Patients and/or the public were not involved in the design, or conduct, or reporting, or dissemination plans of this research.

**Patient consent for publication** Not required.

**Provenance and peer review** Not commissioned; externally peer reviewed.

**Data availability statement** Data are available upon reasonable request. Requests for sharing study data must be made on specific grounds, either (1) with the aim of corroborating the study results in the interest of public health or (2) in the context of an audit by a competent authority. Sufficient information needs to be provided to confirm that the request is made for one of the above-mentioned purposes, including a sound justification and, in case of a request with a view to corroborate study results, a protocol on the research for which the data will be used or a plan for quality control checks, as applicable.

**ORCID iD**
E. Houben http://orcid.org/0000-0002-9173-3335

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
