## [Reviewer comments · BMJ Open]

ARTICLE DETAILS

TITLE (PROVISIONAL)	Cohort Profile: the PHARMO Perinatal Research Network (PPRN) in the Netherlands: a population-based mother-child linked cohort
AUTHORS	Houben, Eline; Broeders, Lisa; Steegers, Eric; Herings, Ron

VERSION 1 – REVIEW

REVIEWER	Olof Stephansson Karolinska Institutet, Sweden
REVIEW RETURNED	26-Mar-2020

GENERAL COMMENTS	Review of manuscript – BMJOpen 2020-037837 Thank you for the opportunity to review this manuscript. This is a manuscript describing the PHARMO Perinatal Research Network in the Netherlands (PPRN). From 1999-2017 some 542,900 births with pregnancy and neonatal data linked to follow-up data for the mothers and infants up to nearly 2 years. The research network include data on demographics, medication use, medical conditions and details concerning birth and neonatal outcomes. In general, the manuscript provides the reader with important information on the PPRN, its strengths and limitations. General comments 1. It would be of value to have an expanded description on the inclusion of births in the cohort. What proportion of all births are included? Could you provide a map of the Netherlands to further illustrate inclusion?2. It would also be helpful with an extended section on the cohort over calendar time. How is the inclusion by calendar year in the Netherlands?3. Furthermore, given the high proportion of home births in the Netherlands, is this cohort based on hospital births only?4. How is missing data handled in the PPRN? What proportion of missing is there in the data?5. What proportion of births were not included in the linkage? Is there no personal registration number for citizens of the Netherlands?6. Could the authors please describe the funding for the project?7. Was there an ethics committee approval before initiating PPRN or do you need it for each study performed?
--

REVIEWER	Elsa Lorthe EPIUnit, ISPUP, Portugal
REVIEW RETURNED	22-Apr-2020

GENERAL COMMENTS	This article presents the main features of the PHARMO Perinatal Research Network (PPRN), which was set up in the Netherlands.
---

	Overall, the manuscript is clear and well-written, and provides information to understand the design of this linked cohort. However, I would be grateful if the authors could consider the following suggestions.  1. Abstract: I think the first sentence is useless for this cohort profile. It is true that pregnant women are often excluded from RCTs for many reasons. However, many RCTs assess specifically the impact of interventions in pregnant women, and under certain conditions, observational studies can provide relevant information, based on a real-life setting and including for interventions that couldn't be evaluated using a RCT design. Thus, I would emphasize the strengths of observational population-based studies instead of presenting such a design as an alternative to the lack of RCTs. 2. Introduction: Same comment than in the abstract. If the authors refer specifically to RCTs assessing medications in pregnant women, they should clarify from the first sentence that it was the original intention behind the PPRN. 3. Setting: In this section, I would expect a description of the population-based extent of this linked cohort. Which are the hospitals participating (private and public)? Are home births included? etc 4. Data sources: Is there any validation of clinical data included in Perined? Do women consent to be part of this registry? 5. Why can PHARMO be considered population-based, as the authors state that "data [come] from different primary and secondary healthcare settings"? I would provide more description, to help the reader understand the design. For instance, I would explain what it the basis for participating in PHARMO: voluntary? Geographically-based? 6. I would also provide a reference (if existing) for the PHARMO validated algorithms on page 3. 7. Data collection (page 4): How many pregnancies could not be linked and were thus excluded? 8. Table 1: It is not clear whether the record types (medication, GP and so on) are available for both the mother and child or only for one of them. 9. Who is in charge of children's medical follow-up in the Netherlands? GPs? Pediatricians (if yes, do they contribute to the PPRN data)? 10. Table 2: Is it possible to link several pregnancies to the same mother? Is it possible to perform studies comparing siblings? 11. Are laboratory tests performed during pregnancy available? 12. Findings: On page 8, line 25, the authors refer to "uncomplicated pregnancy", which doesn't seem appropriate in the context of preterm birth. 13. Findings: I'm surprised to read few details about pharmacoepidemiological findings, which were the primary reason for setting up this linked cohort. Conversely, the results about the comparison of preterm and term babies' outcomes can be mentioned, but do not bring any new information regarding the available scientific literature. Overall, few studies were published using these data. Why? 14. Strengths and limitations: On page 8, the authors state "drug use around pregnancy". This is a bit vague. Do they refer to "during pregnancy" or do include as well the periods before and after pregnancy? 15. I would state earlier in the article that all pregnancy outcomes are included (including TOP and stillbirths).
--	---

	16. Figure 2: Why are there “only” 130,000 children in the cohort? I would include the year of birth in this graph and would explain the reasons for such a loss to (passive) follow-up. 17. I would also provide the direct links to the website page where further information can be retrieved. The Perined website is only available in Dutch, which is tricky for a foreign reader. 18. Was this cohort approved by an ethics committee? If no, why? Are participants informed of the use of their data?
--	--

VERSION 1 – AUTHOR RESPONSE

Reviewer #1:

Comment 1: It would be of value to have an expanded description on the inclusion of births in the cohort. What proportion of all births are included? Could you provide a map of the Netherlands to further illustrate inclusion?

Answer: Thank you for these questions. A PHARMO-Perined linkage could be established for 17% (542,900 out of 3,200,000) of the births included in Perined over the years 1999-2017. Seeing that this is only a subsample of the nationwide Perined registry, we assessed the representativeness of those linked pregnancies compared to all the Perined pregnancies, of which a summary is presented in Table 2. We have now clarified earlier in the text that the PHARMO Database Network concerns subnational data, in contrary to Perined (*“PHARMO is a population-based network of databases combining subnational data from different primary and secondary healthcare settings in the Netherlands. These different data sources, including data from general practices, in- and out-patient pharmacies, clinical laboratories, hospitals, the cancer registry, pathology registry and perinatal registry, are linked on a patient level through validated algorithms.”* – p.4 r.94). Detailed information on the data captured by PHARMO is also available from the paper that was recently published, which we have now provided as a reference (Kuiper et al., *Clin Epidemiol* 2020). Unfortunately, we do not have the level of detail available to be able to provide a map of the Netherlands that represents the database coverage, but we know from the information that is provided by STIZON (Trusted Third Party for database linkages) that the PHARMO catchment area is not restricted to a certain region, but spread all over the country. We have also added an expanded description on the inclusion of births in the cohort in the ‘Data collection’ section (*“Further characterisation of the PPRN is included in Table 2, including the total Perined population as a reference, considering that only a subsample of the Netherlands is represented by the PHARMO Database Network.”* – p.5 r.125).

Comment 2: It would also be helpful with an extended section on the cohort over calendar time. How is the inclusion by calendar year in the Netherlands?

Answer: Thank you for this nice suggestion, we have included an additional figure (Figure 2, see below) presenting the number of pregnancies per calendar year in the two cohorts.

Comment 3: Furthermore, given the high proportion of home births in the Netherlands, is this cohort based on hospital births only?

Answer: No restrictions apply with regard to the place of birth for the PPRN Pregnancy Cohort, so both home and hospital births are included. We have added clarification in the ‘Data sources’ section (*“The established registry reflects virtually all deliveries in the Netherlands (~99% agreement with the Municipal Administration), i.e. including home as well as hospital births.” – p.4 r.90*).

Comment 4: How is missing data handled in the PPRN? What proportion of missing is there in the data?

Answer: As for Perined, reports are sent every month to the care providers who fill in the data. This alerts them to the missing and invalid items. They are also approached every quarter if the delivery is behind schedule compared with last year. Perined does not impute missing values, everything must be entered/changed in the source of the data. For most of the core variables in Perined, the percentage missing is below 3%. As for the number of births included in Perined, this is about 99% of all births in the Netherlands. For databases from the PHARMO Database Network, similar methods apply. Upon receipt, completeness of the data is checked by STIZON and healthcare providers are asked to update their systems in case required variables are missing, before the data are transferred to the PHARMO Database Network. Any remaining missing values are presented as ‘unknown’ in the results. For analyses that require completeness of records, imputation of missing values is sometimes performed and will then be described as such in the methods section. As the data are not primarily collected for research purposes, it is limited to the data that are captured as part of the medical records or other linked data sources in daily practice and therefore we will always have to deal with some missing values. The proportion missing is depending on many different factors: the variable, the database(s), the patient population and therefore no overall proportion of missing can be provided for the PHARMO Database Network. We have updated the ‘Strengths and limitations’ section with this information (*“As with any database, identification of medical events is limited to data that are captured as part of the medical records or other linked data sources in daily clinical practice. These data are not primarily collected for research purposes and rely on appropriate diagnostic coding.” – p.10 r.180*).

Comment 5: What proportion of births were not included in the linkage? Is there no personal registration number for citizens of the Netherlands?

Answer: In total, a PHARMO-Perined linkage could be established for 17% (542,900 out of 3,200,000) of the births included in Perined over the years 1999-2017. This means that approximately 4 out of 5 births were not included in the linkage. Citizens of the Netherlands indeed all have a social security number, however as described in the 'Strengths and limitations' section the number of records that includes this number is currently too limited to allow for deterministic record linkage between PHARMO and Perined.

Comment 6: Could the authors please describe the funding for the project?

Answer: For the current study presenting the characteristics of the PPRN, no specific grant from any funding agency in the public, commercial, or not-for-profit sectors was received. As we believe it is appropriate to also mention the funding for the first study in which the PHARMO-Perined linkage algorithms were developed, we have slightly updated this section in the paper ("*The first study in which the linkage algorithms between PHARMO and Perined were developed was funded by Abbott.*" – p.13 r.230).

Comment 7: Was there an ethics committee approval before initiating PPRN or do you need it for each study performed?

Answer: Studies performed using the PPRN concern database research with anonymous data and therefore no Institutional Review Board or ethics committee approval is required. Each separate study requires permission of the applicable compliance and privacy boards of both PHARMO and Perined. We have expanded this in the 'Collaboration' section of the paper for clarification ("*As it concerns database research with anonymous data, no Institutional Review Board or ethics committee approval is required.*" – p.11 r.207).

Reviewer #2:

Comment 1: Abstract: I think the first sentence is useless for this cohort profile. It is true that pregnant women are often excluded from RCTs for many reasons. However, many RCTs assess specifically the impact of interventions in pregnant women, and under certain conditions, observational studies can provide relevant information, based on a real-life setting and including for interventions that couldn't be evaluated using a RCT design. Thus, I would emphasize the strengths of observational population-based studies instead of presenting such a design as an alternative to the lack of RCTs.

Answer: Thank you for this notification, having evaluated this sentence again we indeed agree with you that it was not completely reflecting what it should. We have therefore updated this sentence, now emphasizing the suitability of observational population-based research in these vulnerable populations ("*Observational population-based research is a very suitable non-invasive method for studies in the vulnerable populations of pregnant women and children.*" – p.2 r.22).

Comment 2: Introduction: Same comment than in the abstract. If the authors refer specifically to RCTs assessing medications in pregnant women, they should clarify from the first sentence that it was the original intention behind the PPRN.

Answer: See previous comment – we agree that this was not completely reflecting what it should and have therefore updated this sentence ("*Observational population-based research is a very suitable non-invasive method for studies in the vulnerable populations of pregnant women and children.*" – p.4 r.63).

Comment 3: Setting: In this section, I would expect a description of the population-based extent of this linked cohort. Which are the hospitals participating (private and public)? Are home births included? etc

Answer: Thank you for these questions. We understand your request for emphasis on the population-based extent and have therefore slightly updated this section ("*With data collection starting in 1999, the linkage of these population-based data sources facilitates large-scaled observational pharmacoepidemiological perinatal research.*"). We felt that further description of the population-based nature of the data sources would better fit in the 'Data sources' section, where we have added clarification that home as well as hospital births are captured ("*The established registry reflects virtually all*

deliveries in the Netherlands (~99% agreement with the Municipal Administration), i.e. including home as well as hospital births.” – p.4 r.90). No relevant distinction between private and public hospitals applies, because Perined captures virtually all deliveries in the Netherlands. As PHARMO is not a nationwide database in contrast to Perined, we have updated this section with regard to representativeness (“PHARMO covers approximately a quarter of the Dutch population and is shown to be representative of the Dutch population with regard to age and sex; however data collection period, catchment area and overlap between data sources differ.” – p.5 r.106). Further details on the build-up of PHARMO can be found in the paper that was recently published for which we added a reference (Kuiper et al., *Clin Epidemiol* 2020).

Comment 4: Data sources: Is there any validation of clinical data included in Perined? Do women consent to be part of this registry?

Answer: The data is validated by the reports sent every month to the care providers. They need to check the warnings displayed in the report.

The perinatal registration offers the possibility to process patient data as part of their medical file, in the context of measuring the quality of care. This means that it is seen as part of the patient file in the context of the WGBO (Dutch law: Medical Treatment Contract Act (“Wet op de geneeskundige behandelingsovereenkomst”). This is done with assumed permission. Research is also being conducted with the data. In principle, this does not contain traceable medical data. If it is nevertheless necessary for certain research to work with traceable data, explicit permission must be requested for this.

Comment 5: Why can PHARMO be considered population-based, as the authors state that “data [come] from different primary and secondary healthcare settings”? I would provide more description, to help the reader understand the design. For instance, I would explain what it the basis for participating in PHARMO: voluntary? Geographically-based?

Answer: PHARMO can be regarded population-based as it is a registry that contains records for people diagnosed with a specific type of disease or who are treated with certain medication who reside within a defined geographic region, without it being limited to hospitalised patients. There is a voluntary basis for participation in PHARMO, without a pre-defined geographical restriction, but data being collected from all over the country. Data are retrieved directly from the source, i.e. the electronic medical records of the healthcare providers who agree to contribute to the PHARMO Database Network. All patients registered at the contributing healthcare providers are included, unless the patient requested to opt out. We have provided this description in the ‘Data sources’ section (“Data are retrieved directly from the source, i.e. the electronic medical records of the healthcare providers who agree to contribute to the PHARMO Database Network. All patients registered at the contributing healthcare providers are included, unless the patient requested to opt out.” – p.5 r.98), together with a reference to the paper that was recently published describing also the data collection process (Kuiper et al., *Clin Epidemiol* 2020).

Comment 6: I would also provide a reference (if existing) for the PHARMO validated algorithms on page 3.

Answer: We agree that this needs proper citation, which is why we refer to the detailed information on the methodology and the validation of the used record linkage method further down this section (“Detailed information on the methodology and the validation of the used record linkage method can be found elsewhere.” – p.5 r.105). Also, we have added a reference to the paper describing the PHARMO Database Network (including its linkages) that was recently published (Kuiper et al., *Clin Epidemiol* 2020).

Comment 7: Data collection (page 4): How many pregnancies could not be linked and were thus excluded?

Answer: In total, a PHARMO-Perined linkage could be established for 17% (542,900 out of 3,200,000) of the births included in Perined over the years 1999-2017. This means that approximately 4 out of 5 births were not included in the linkage. Seeing that this linkage could only be established for a subsample of the nationwide Perined registry, we assessed the representativeness of those linked pregnancies compared to all the Perined pregnancies, of which a summary is presented in Table 2.

Comment 8: Table 1: It is not clear whether the record types (medication, GP and so on) are available for both the mother and child or only for one of them.

Answer: Thank you for this relevant question. The presented record types are available for both the mother and the child. We have added clarification for this in the 'Data collection' section, as we agree that this was somewhat unclear ("*A schematic overview of data captured in the PPRN for mothers and children and how these two cohorts interrelate is included in Figure 1 and Table 1.*" – p.5 r.123). As noted also in the 'Strengths and limitations' section, the data collection periods and catchment areas between the different databases vary and therefore availability differs per patient, depending on the databases included.

Comment 9: Who is in charge of children's medical follow-up in the Netherlands? GPs? Pediatricians (if yes, do they contribute to the PPRN data?)?

Answer: In the Netherlands, the GP acts as the gatekeeper of the healthcare system and all patients, regardless of age or insurance, are uniquely registered by a single GP. Access to secondary care is only through referral by the GP and medical information from secondary care is communicated back to the GP for inclusion in the medical files. This information will be captured in the GP Database part of the PPRN. Detailed information from secondary care (e.g. by a pediatrician) will not be captured in this GP Database, but can be retrieved (for a subcohort) from the Hospital Database part of the PPRN, capturing both in-patient (i.e. hospital admissions) as well as out-patient (i.e. ambulatory care) records. Information on care provided by the pediatrician directly after birth is captured in Perined, also part of the PPRN data. We have updated Table 1 to illustrate this (rows "General practitioner" and "Ambulatory care") and to emphasize relevant data sources for this pediatric population.

Comment 10: Table 2: Is it possible to link several pregnancies to the same mother? Is it possible to perform studies comparing siblings?

Answer: Yes, pregnancies can be linked to the same mother based on a unique identifier that exists in the data for every mother. This way, it is also possible to perform studies comparing siblings. However, it should be noted (see section 'Strengths and limitations') that multiple birth pregnancies are currently underrepresented in the Child Cohort, because the number of available deterministic characteristics is often too limited to differentiate between the siblings of a multiple birth. Comparisons can more easily be made for siblings of singleton pregnancies.

Comment 11: Are laboratory tests performed during pregnancy available?

Answer: Yes, laboratory tests performed during pregnancy are available for a subcohort of the pregnancies included in the PPRN. They can be extracted from the Clinical Laboratory Database (for tests requested by general practitioners or medical specialists) and/or from the GP Database (for tests requested by general practitioners).

Comment 12: Findings: On page 8, line 25, the authors refer to "uncomplicated pregnancy", which doesn't seem appropriate in the context of preterm birth.

Answer: Thank you for this notification. Although this terminology is taken directly from the publication, I agree with you that this wording somehow is confusing and not completely appropriate. It was meant to refer to preterm birth following an uncomplicated pregnancy. However, we have updated the text and deleted the 'uncomplicated pregnancy' part as it was not really relevant to the message we wanted to bring ("*Similar long-term comparisons of morbidities and health-care utilisation have been made which showed that children born following spontaneous preterm labour (irrespective of gestational age at delivery) were at increased risk of neurodevelopmental and respiratory conditions compared with those from full-term labour pregnancies.*" – p.10 r.156).

Comment 13: Findings: I'm surprised to read few details about pharmacoepidemiological findings, which were the primary reason for setting up this linked cohort. Conversely, the results about the comparison of preterm and term babies' outcomes can be mentioned, but do

not bring any new information regarding the available scientific literature. Overall, few studies were published using these data. Why?

Answer: Indeed, the number of published studies using these data is, thus far, still fairly limited. The reason for this is that the linkage of these two data sources is quite new. Many of the studies we have performed over the last years have just recently been finalized and are currently in the process of manuscript development, or studies still need to be finalized. For some studies no scientific paper has been prepared for publishing, but only a scientific report. By means of the current paper we want to notify a broader audience about its existence and applicability for setting up future studies in this field, for which we believe this is an invaluable resource.

Comment 14: Strengths and limitations: On page 8, the authors state “drug use around pregnancy”. This is a bit vague. Do they refer to “during pregnancy” or do include as well the periods before and after pregnancy?

Answer: Thank you for this relevant question. We meant to refer to the pregnancy period as well as the periods before and after pregnancy. We agree with you that the terminology ‘around’ was a bit vague, which is why we have revised this sentence (“*The 542,900 pregnancies linked in the data cut up to 2017 allow for assessment of drug use during the 9-month preconception, pregnancy and 9-month postpartum periods.*” – p. 10 r. 178).

Comment 15: I would state earlier in the article that all pregnancy outcomes are included (including TOP and stillbirths).

Answer: Thank you for this suggestion, we have now stated this already in the ‘Data sources’ section (“*Perined is a nationwide registry in which medical data around pregnancy and birth is included from pregnancies with a gestational age of at least 16 weeks (including terminated pregnancies and stillborns).*” – p. 4 r. 80).

Comment 16: Figure 2: Why are there “only” 130,000 children in the cohort? I would include the year of birth in this graph and would explain the reasons for such a loss to (passive) follow-up.

Answer: The children presented in this figure are those included in the ‘Child Cohort’, which exists of those pregnancies for which an individual mother-child linkage could be established (currently available for about a quarter of the pregnancies included in the ‘Pregnancy Cohort’) (see also section ‘Data collection’). These 126,200 pregnancies are also represented in Table 2. We did some experimenting in trying to fit in the birth years in this graph, however this did not provide a clear figure and therefore we have added these numbers to a newly created Figure 2 (previous Figure 2 is now Figure 3). Furthermore, we have added the possible reasons for a child being lost to follow-up at the end of the section ‘Data collection’ (“*Details on the available database follow-up for the children included in the Child Cohort are presented in Figure 3, with end of follow-up defined by either end of database registration (i.e. patient moves out of the PHARMO catchment area), death or end of study period (December 31st, 2018), whichever occurred first.*” – p. 5 r. 128).

Comment 17: I would also provide the direct links to the website page where further information can be retrieved. The Perined website is only available in Dutch, which is tricky for a foreign reader.

Answer: Thanks for this suggestion, we have now provided the links that give direct information on the data requests (“*An overview of the variables included in the different databases, the terms and conditions and data application forms are available on www.pharmo.nl/what-we-have/data-request-PHARMO and (in Dutch) www.perined.nl/registratie/faciliteren-onderzoek.*” – p. 11 r. 208). Unfortunately the Perined website is indeed only available in Dutch, which we have now shortly mentioned also in this section.

Comment 18: Was this cohort approved by an ethics committee? If no, why? Are participants informed of the use of their data?

Answer: As it concerns database research with anonymous data, no Institutional Review Board or ethics committee approval is required. We have clarified this in the ‘Collaboration’ section (“*As it concerns database research with anonymous data, no Institutional*

Review Board or ethics committee approval is required.” – p.11 r.207). The data is handled in accordance with data protection, privacy regulations and ISO certification schemes. Each data request is checked against these policies and requires permission of the applicable compliance and privacy boards of both PHARMO and Perined. Participants are not separately informed about the use of their data.

VERSION 2 – REVIEW

REVIEWER	Elsa Lorthe EPIUnit, ISPUP, Portugal
REVIEW RETURNED	10-Jul-2020

GENERAL COMMENTS	I would like to thank the authors for providing a revised version of their article. Although I would have appreciated a point-by-point response, I think that my comments have been adequately addressed.
---